

# Attitudes towards preventive tuberculosis treatment among hospital staff

Vidya Pathak[1,2], Zinta Harrington[1,2] and Claudia C. Dobler[1,2]

[1] South Western Sydney Clinical School, University of New South Wales, Sydney, New South Wales, Australia
[2] Department of Respiratory Medicine, Liverpool Hospital, Sydney, New South Wales, Australia

## ABSTRACT

**Background.** Healthcare workers have an increased risk of latent tuberculosis infection (LTBI), but previous studies suggested that they might be reluctant to accept preventive tuberculosis (TB) treatment. We aimed to examine doctors' and nurses' experience of TB screening and to explore their attitudes towards preventive TB treatment.

**Methods.** We conducted a survey among randomly selected healthcare workers at a tertiary hospital in Sydney, Australia, using a paper-based questionnaire.

**Results.** A total of 1,304 questionnaires were distributed and 311 (24%) responses were received. The majority of hospital staff supported preventive TB treatment in health care workers with evidence of latent TB infection (LTBI) in general (74%, 164/223) and for them personally (81%, 198/244) while 80 and 53 healthcare workers respectively had no opinion on the topic. Staff working in respiratory medicine were significantly less likely to support preventive TB treatment in health care workers in general or for them personally if they would have evidence of LTBI compared to other specialties ($p = 0.001$). Only 13% (14/106) of respondents with evidence of LTBI indicated that they had been offered preventive TB treatment. Twenty-one percent (64/306) of respondents indicated that they did not know the difference between active and latent TB. Among staff who had undergone testing for LTBI, only 33% (75/230) felt adequately informed about the meaning of their test results.

**Discussion.** Hospital staff in general had positive attitudes towards preventive TB treatment, but actual treatment rates were low and perceived knowledge about LTBI was insufficient among a significant proportion of staff. The gap between high support for preventive TB treatment among staff and low treatment rates needs to be addressed. Better education on the concept of LTBI and the meaning of screening test results is required.

## INTRODUCTION

Australia has a low incidence of tuberculosis (TB), with an incidence of 6.0 per 100,000 population per year (*Barry et al., 2012*). However, ongoing TB control efforts are warranted given high levels of immigration from settings with a high incidence of TB. Also, a significant number of migrants from TB endemic areas work in the Australian health system. Healthcare workers (HCWs) have been shown to be at an increased risk

Corresponding author
Claudia C. Dobler,
c.dobler@unsw.edu.au

of having latent (dormant) tuberculosis infection (LTBI). The Melbourne Mantoux Study that was conducted in the late 1990s showed that 19.3% of HCWs in teaching hospitals in Melbourne had positive tuberculin skin tests (TSTs), compared to 13.7% of non-HCWs ($p < 0.001$) (*Stuart et al., 2001*). The risk of developing active TB disease from LTBI is of concern in HCWs, as sick HCWs could pass the TB infection on to vulnerable patients. People with LTBI can receive preventive TB treatment, which usually consists of a 6–9 month course of daily isoniazid tablets. However, this treatment is associated with a potential risk of drug-induced hepatitis, which varies from 0.1 to 2.4% (depending on age) for hepatitis requiring hospitalisation (*Smith et al., 2011*).

While the efficacy of isoniazid preventive therapy has been clearly demonstrated, (*Ayieko et al., 2014*; *Smieja et al., 2000*; *World Health Organization, 1982*) HCWs have been shown to be significantly more reluctant to accept treatment for LTBI than non-HCWs (*Barrett-Connor, 1979*; *Camins et al., 1996*; *Geiseler, Nelson & Crispen, 1987*; *Gershon et al., 2004*; *LoBue & Catanzaro, 1998*; *Ramphal-Naley et al., 1996*; *Xu & Schwartzman, 2010*). HCWs have lower rates of initiation of treatment for LTBI and may face specific issues when undergoing screening for LTBI, such as scepticism towards medical testing and difficulties scheduling a screening appointment when doing shift work (*Joseph et al., 2004*; *Ramphal-Naley et al., 1996*). The exact reasons for lower uptake of this evidence-based intervention among HCWs remain unclear.

The aim of this study was to examine HCWs experiences of TB screening and their attitudes towards preventive TB treatment at a tertiary hospital in Sydney, Australia.

## MATERIALS & METHODS

### Study setting and population

The study was performed at Liverpool Hospital, a tertiary hospital with a major TB clinic in Sydney. Liverpool Hospital TB clinic treats approximately 10% of all TB patients in Australia and 20% of all patients in the state of New South Wales (NSW). In this setting, the TB incidence is 7.3 per 100,000 population per year (*Barry et al., 2012*).

Since the introduction of a new policy directive on HCW screening in NSW in 2011 new HCWs must only undergo TB screening if they were born in a country with a high incidence of TB, or if they have travelled to a country with a high incidence of TB for three months or longer (*NSW Health, 2011*). The state policy directive defines a country with a high incidence of TB as one that has a TB incidence ≥60 cases per 100,000 population per year. Existing staff members who lived in a country with a high incidence of TB for more than 3 months within the last 3 years must also undergo TB screening.

During the screening process, a medical history is taken; a tuberculin skin test (TST) is performed and a chest X-ray is done if the TST is positive. The results of these investigations are then reviewed by a doctor at the TB clinic, who decides whether the HCW is booked for a doctor's appointment or not. When there is no evidence of active TB, the doctor will decide whether to offer preventive treatment to a HCW with evidence of LTBI (usually TST ≥ 10 mm) based on an assessment of the individual's risk of developing TB. If there is concern about a possible false positive TST result (e.g., in a HCW who

received a Bacillus Calmette-Guérin (BCG) vaccination, which is known to cause a TST cross-reaction), the doctor may also decide to order an additional blood test (Interferon Gamma Release Assay (IGRA)) to confirm true LTBI.

### Study design

A paper-based questionnaire was sent to randomly selected nurses and doctors, attached to the fortnightly pay slips. Respondents were asked to return the completed questionnaire anonymously in the attached return envelope. Randomisation was performed using a random number generator in Microsoft Excel. Additionally, one of the researchers (VP) approached doctors and nurses in person using a random starting point from a list of hospital locations (including wards, outpatient clinics and clinical meetings) and then approaching every second HCW. Staff who were approached in person were asked if they had received a questionnaire with the pay slip. If they indicated that they had not yet received a questionnaire, they were given a questionnaire and asked to return it anonymously in the attached envelope. They were asked to fill in only one questionnaire should they later discover a mailed questionnaire at home. We did not have a formal control mechanism to exclude duplicate submissions from the same person. The risk for duplicate submissions was, however, perceived to be very low because there was no financial incentive attached to providing a response.

### Questionnaire

The questionnaire for hospital staff (see supporting information) included questions on HCWs' demographic background including their age, gender, profession, hospital department, level of patient contact, country of birth, and history of overseas stay. Additional questions addressed the personal experience with TB screening including history of BCG vaccination, results of screening chest X-rays, results of TST and IGRA tests and knowledge about their personal TB status (no evidence of LTBI, evidence of LTBI, active TB or past medical history of TB). Participants were asked whether they were offered treatment for LTBI and, whether they accepted treatment for LTBI. They were questioned about their general opinion on treatment for LTBI.

### Definitions

A TB incidence of ≥60 per 100,000 population was chosen to define a country with a high incidence of TB, as per the NSW Health Policy Directive (*NSW Health, 2011*). Incidence rates of TB were based on estimates of the World Health Organization (WHO) accessed on 01/10/2013 (*World Health Organization, 2013*). A positive screening test result was defined as having a positive TST (≥10 mm) and/or IGRA. Staff working in respiratory medicine was defined as HCWs whose work focused on the care of patients with respiratory problems, not just staff involved in the care of TB patients. The speciality of respiratory medicine was highlighted in the study because the TB clinic at the study hospital was integrated into the department of respiratory medicine.

### Statistical analyses

Calculation of proportion of responses was based on the total number of HCWs who answered a specific question in the questionnaire, in other words, the denominator for

each question was the number of staff who answered the question by ticking one of the boxes. Data were extracted from the paper-based questionnaire and entered into a Microsoft Excel database.

Logistic regression and odds ratios (ORs) with 95% confidence intervals (95% CIs) were used to assess associations between demographic characteristics and survey responses. The independent effect of potential predictors of survey responses among HCWs was estimated using multivariable logistic regression for binary outcomes, in which the dependent variable could take only two values, yes/no. Responses that indicated that the respondent had no opinion or did not know the answer to the question were thus excluded for this analysis. Multivariable analysis included adjustment for age, sex, profession, specialty, country of birth, screening results, being offered treatment and history of BCG vaccine. Statistically significant results were defined as $p < 0.05$ SPSS Statistics v21 (IBM Corp, Armonk, NY, USA) was used for the statistical analysis.

### Ethical approval & consent

The study protocol was approved by the South Western Sydney Local Health District Ethics Committee (project number 12/265). Attached to all questionnaires was a participant information sheet which included the following information regarding participant consent: "Your participation in this survey is entirely voluntary. You may choose not to take the survey, or to skip any questions that you do not want to answer. Your completion of the questionnaire serves as your voluntary agreement to participate in this research project."

## RESULTS

### Participants

A total of 733 questionnaires were sent to randomly selected nurses and 431 to randomly selected doctors; an additional 87 and 53 questionnaires were directly handed to nurses and doctors respectively. Of 1,304 HCWs who received a questionnaire, 318 responded, resulting in a response rate of 24%. Seven responses were excluded because the questionnaires were blank, so 311 responses were included in the analysis. The response rate was 22% (105/484) among doctors and 25% (206/820) among nurses. Forty-seven percent (144/307) of respondents were Australian-born. Ninety-five percent of respondents (295/311) had undergone TB screening, including 151who had undergone screening at the hospital's TB clinic. The majority (225) had undergone screening prior to February 2011, when a new screening policy for HCWs was introduced in NSW, limiting TB screening to HCWs considered to be at risk of previous TB infection. Table 1 summarizes the demographic characteristics of all HCWs who participated in the study.

### Experience with screening for latent TB infection at the hospitals' TB clinic

Of the 295 HCWs who underwent screening, 258 (87%) reported having undergone TST and/or IGRA testing (Fig. 1). Of those who underwent a TST and/or IGRA test, 116 (45%) indicated that they had a positive screening test result suggestive of LTBI, of which 10

**Table 1  Characteristics of HCWs who participated in the survey.**

| Characteristic | Total respondents $n = 311$ $n$(%) |
|---|---|
| Age | |
| <30 | 75 (24%) |
| 31–40 | 96 (31%) |
| 41–50 | 78 (25%) |
| ≥50 | 62 (20%) |
| Sex | |
| Male | 90 (29%) |
| Female | 221 (71%) |
| Profession | |
| Doctor | 105 (34%) |
| Nurse | 206 (66%) |
| Department ($n = 309$)[a] | |
| Respiratory | 28 (9%) |
| Other medical | 131 (42%) |
| Surgery | 36 (12%) |
| Other | 114 (37%) |
| Country of Birth ($n = 307$)[a] | |
| Australia | 144 (47%) |
| Overseas from a country with a TB incidence <60 per 100,000 population per year | 64 (21%) |
| Overseas from a country with a TB incidence ≥60 per 100,000 population per year | 99 (32%) |

**Notes.**

[a] Based on the number of respondents who answered this question.

(9%) did not answer the question on whether they were offered treatment for LTBI or not. Only 13% (14/106) of HCWs with evidence of LTBI indicated that they were offered preventive TB treatment; 40% (42/106) were not offered treatment, and 47% (50/106) felt that this question did not apply to them. Sixty-four percent of HCWs (9/14) accepted the offered preventive TB treatment. HCWs who refused treatment felt that they could monitor signs and symptoms of TB themselves (3/5); stated they feared side effects (2/5), felt that the treatment was unnecessary (2/5) and stated that they did not need treatment as they had been vaccinated (1/5).

## Attitudes towards preventive TB treatment

Of the respondents who expressed an opinion on the topic, 74% (164/223) supported the idea that hospital staff who had evidence of LTBI should receive preventive treatment while 26% (59/223) did not think that preventive TB treatment was indicated in this situation. A further 80 respondents did not express any opinion on the topic. In bivariate logistic regression analysis there was a positive association between support for preventive TB treatment for employees in non-respiratory specialties (OR 7.83, 95% CI [2.27–22.51] for staff in medical specialties other than respiratory medicine and OR 20.83 95% CI [4.53–95.89] for surgical staff, compared to respiratory staff). There were also associations

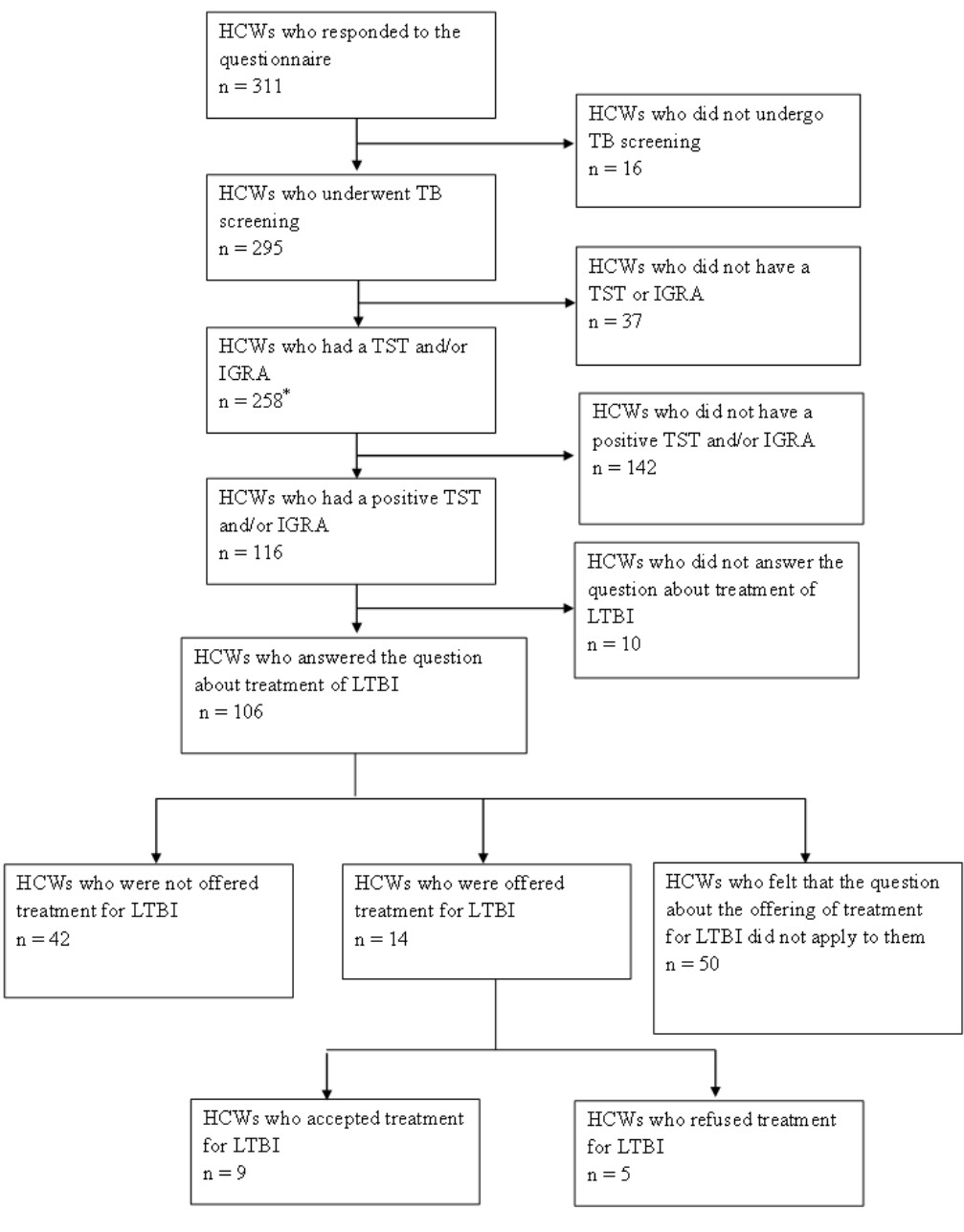

*Sixteen had both a TST and IGRA

**Figure 1 Summary of respondents' TB screening results.**

between support for preventive TB treatment and female gender (OR 1.94, 95% CI [1.05–3.61]), nursing staff (OR 2.36, 95% CI [1.28–4.34]) and positive TST status (OR 2.41 95% CI [1.21–4.82]). In multivariable logistic regression, only the respondent's specialty had a significant association with a favourable attitude towards preventive TB treatment. Staff from non-respiratory specialties were more supportive of preventive TB treatment

**Table 2 Attitudes towards LTBI treatment among respondents by specialty.[a]**

| Question | Yes (%) | No (%) | Unadjusted OR (95% CI) | p value | Adjusted OR (95% CI) | Adjusted p value |
|---|---|---|---|---|---|---|
| Do you think staff who have evidence of LTBI should receive preventive treatment? | | | | <0.001[*] | | 0.001[*] |
| All | 164/223 (74%) | 59/223 (26%) | | | | |
| Respiratory | 6/21 (29%) | 15/21 (71%) | 1.00 (Ref) | | 1.00 (Ref) | |
| Other medical | 72/95 (76%) | 23/95 (24%) | 7.83 (2.72–22.52) | <0.001[*] | 8.62 (2.69–27.61) | <0.001[*] |
| Surgery | 25/28 (89%) | 3/28 (11%) | 20.83 (4.53–95.89) | <0.001[*] | 16.94 (3.22–89.05) | 0.001[*] |
| Other | 61/79 (77%) | 18/79 (23%) | 8.47 (2.87–25.02) | <0.001[*] | 7.05 (2.19–22.72) | 0.001[*] |
| If you had LTBI, would you like to be offered preventive therapy? | | | | <0.001[*] | | 0.001[*] |
| All | 198/244 (81%) | 46/244 (19%) | | | | |
| Respiratory | 11/23 (48%) | 12/23 (52%) | 1.00 (Ref) | | 1.00 (Ref) | |
| Other medical | 96/108 (89%) | 12/108 (11%) | 8.73 (3.16–24.07) | <0.001[*] | 10.16 (3.29–31.44) | <0.001[*] |
| Surgery | 24/27 (89%) | 3/27 (11%) | 8.73 (2.04–37.30) | 0.003[*] | 10.56 (2.11–52.98) | 0.004[*] |
| Other | 67/86 (78%) | 19/86 (22%) | 3.85 (1.47–10.09) | 0.006[*] | 3.97 (1.34–11.71) | 0.013[*] |
| Do you feel you need more information before deciding to accept or reject preventive therapy? | | | | 0.198 | | 0.397 |
| All | 185/264 (70%) | 79/264 (30%) | | | | |
| Respiratory | 12/25 (48%) | 13/25 (52%) | 1.00 (Ref) | | 1.00 (Ref) | |
| Other medical | 80/112 (71%) | 32/112 (29%) | 2.71 (1.12–6.56) | 0.027[*] | 2.68 (1.00–7.23) | 0.051 |
| Surgery | 22/30 (73%) | 8/30 (27%) | 2.98 (0.97–9.20) | 0.058 | 2.13 (0.61–7.50) | 0.239 |
| Other | 70/96 (73%) | 26/96 (27%) | 2.92 (1.18–7.21) | 0.020[*] | 2.49 (0.91–6.83) | 0.076 |

**Notes.**
[a]Responses that indicated that the person filling in the questionnaire had no opinion on the topic were excluded for this analysis.
[*]Denotes statistical significance.

than respiratory staff (aOR 8.62, 95% CI [2.69–27.61] for staff in other medical specialties and aOR 16.94 95% CI [3.22–89.05] for surgical staff) (Table 2).

Eighty-one percent (198/244) of respondents were in favour of being offered preventive TB treatment if they had evidence of LTBI personally and 19% (46/244) preferred not be offered preventive TB treatment personally. Another 53 respondents had no opinion on the topic. There was greater support for personal preventive TB treatment among staff from non-respiratory staff compared to respiratory staff (85%, 187/221 versus 48%, 11/23 respectively (Table 2)). The specialty hospital staff worked in was the only characteristic that was significantly associated with the attitude towards personal preventive TB treatment in bivariate and multivariable logistic analyses.

The majority of respondents (70%, 185/264) stated they would require more information to decide whether they personally would want to receive preventive TB treatment if they had evidence of LTBI, 30% (79/264) felt that they were sufficiently informed and a further 31 had no opinion on the topic.

The majority of respondents who expressed an opinion (73%, 200/273) thought that Australian born HCWs without significant overseas stay should be screened for LTBI; 26% (73/273) thought that they should not be screened. A further 34 respondents voiced no opinion. A third of respiratory staff (9/27) believed that these HCWs should be screened, compared to 78% (191/246) of those in other specialities.

### Education and knowledge

Twenty-one percent of respondents (64/306) indicated that they did not know the difference between active and latent TB. Only 69% of respondents aged 30 years or less thought that they knew the difference compared to 89% of those in the 31–40 years age group (aOR 3.32, 95% CI [1.23–8.99]) (Table 3). Nurses were less likely to indicate that they knew the difference than doctors (69%, 138/201 versus 99%, 104/105; aOR 0.033, 95% CI [0.004–0.26]). Univariate analysis revealed that women were significantly more likely than men to indicate that they did not know the difference between the two conditions, but this was not confirmed in multivariable analysis (Table 3).

Of the HCWs who had undergone TST and/or IGRA testing only 33% (75/230) felt adequately informed about the meaning of their test results, 16% (37/230) did not feel adequately informed, and 51% (118/230) were not even aware that the question on information about test results applied to them.

## DISCUSSION

This study, conducted at a tertiary hospital with one of the biggest TB clinics in Australia, found that hospital staff in general had positive attitudes towards preventive TB treatment, but the proportion of staff with evidence of LTBI who had been offered preventive TB treatment was low and knowledge about LTBI and the meaning of test results was insufficient.

In the current study, only 13% (14/106) of HCWs with evidence of LTBI indicated that they were offered preventive TB treatment. The proportion of treatment of LTBI in HCWs in other studies has ranged from 29% to 98.4% (*Camins et al., 1996*; *LoBue & Catanzaro, 1998*; *Xu & Schwartzman, 2010*). The high treatment rate of 98.4% was from a study that focused on HCWs who had a TST conversion and thus were at a relatively high risk of TB reactivation (*Camins et al., 1996*) while our study investigated routine HCW screening. Further, the studies with higher proportions of preventive TB treatment among HCWs were conducted in North America, (*Camins et al., 1996*; *LoBue & Catanzaro, 1998*; *Shukla et al., 2002*; *Xu & Schwartzman, 2010*) where treatment of LTBI is pursued more proactively, mainly as a result of policies directed at TB elimination in the USA. Two of these studies outlined that preventive TB treatment was routinely offered to HCWs with evidence of LTBI, contributing to relatively high treatment rates (*Camins et al., 1996*; *Shukla et al., 2002*).

In our study 9/14 (64%) of staff who were offered preventive TB treatment accepted it. Studies have shown proportions ranging from to 37.5% to 85% for accepting treatment (*Bhanot et al., 2012*; *Camins et al., 1996*; *LoBue & Catanzaro, 1998*; *Ramphal-Naley et al., 1996*; *Shukla et al., 2002*; *Xu & Schwartzman, 2010*). The fact that the proportion of HCWs
**Table 3  Characteristics of HCWs who knew and did not know the difference between active and latent TB.**

| Factor | HCWs who knew the difference between active and latent TB (%) | HCWs who did not know the difference between active and latent TB (%) | Unadjusted OR (95% CI) | p value | Adjusted OR (95% CI) | Adjusted p value |
|---|---|---|---|---|---|---|
| All | 242/306 (79%) | 64/306 (21%) | | | | |
| Age, n = 306 | | | | 0.006* | | 0.034* |
| ≤30 | 51/74 (69%) | 23/74 (31%) | 1.00 (Ref) | | 1.0 | |
| 31–40 | 85/95 (89%) | 10/95 (11%) | 3.83 (1.69–8.70) | 0.001* | 3.32 (1.23–8.99) | 0.008* |
| 41–50 | 63/77 (82%) | 14/77 (18%) | 2.03 (0.95–4.34) | 0.068 | 1.85 (0.73–4.70) | 0.193 |
| >50 | 43/60 (72%) | 17/60 (28%) | 1.14 (0.54–2.41) | 0.730 | 0.85 (0.33–2.19) | 0.735 |
| Sex, n = 306 | | | | 0.001* | | 0.222 |
| Male | 81/88 (92%) | 7/88 (8%) | 1.00 (Ref) | | 1.0 | |
| Female | 161/218 (74%) | 57/218 (26%) | 0.24 (0.11–0.56) | | 0.52 (0.18–1.48) | |
| Profession, n = 306 | | | | <0.001* | | 0.001* |
| Doctor | 104/105 (99%) | 1/105 (1%) | 1.00 (Ref) | | 1.0 | |
| Nurse | 138/201 (69%) | 63/201 (31%) | 0.02 (0.003–0.15) | | 0.033 (0.004–0.26) | |
| Specialty, n = 304 | | | | 0.060 | | 0.699 |
| Respiratory | 25/28 (89%) | 3/28 (11%) | 1.00 (Ref) | | 1.00 (Ref) | |
| Other medical | 109/129 (84%) | 20/129 (16%) | 0.65 (0.18–2.37) | 0.518 | 0.62 (0.15–2.61) | 0.511 |
| Surgery | 25/35 (71%) | 10/35 (29%) | 0.3 (0.07–1.22) | 0.093 | 0.59 (0.13–2.73) | 0.498 |
| Other | 82/112 (73%) | 30/112 (27%) | 0.33 (0.09–1.17) | 0.085 | 0.44 (0.11–1.76) | 0.246 |
| Country of Birth, n = 302 | | | | 0.229 | | 0.799 |
| Australia | 106/142 (75%) | 36/142 (25%) | 1.00 (Ref) | | 1.00 (Ref) | |
| Outside Australia with TB incidence <60 per 100,000 per year | 50/62 (81%) | 12/62 (19%) | 1.42 (0.68–2.95) | 0.354 | 0.65 (0.26–1.63) | 0.363 |
| Outside Australia with TB incidence ≥60 per 100,000 per year | 82/98 (84%) | 16/98 (16%) | 1.74 (0.90–3.35) | 0.098 | 1.03 (0.43–2.59) | 0.940 |
| BCG Vaccine, n = 300 | | | | 0.088 | | 0.163 |
| Yes | 184/227 (81%) | 43/227 (19%) | 0.84 (0.37–1.91) | 0.669 | 0.69 (0.25–1.92) | 0.471 |
| No | 41/49 (84%) | 8/49 (16%) | 1.00 (Ref) | | 1.00 (Ref) | |
| Don't know | 15/24 (63%) | 9/24 (37%) | 0.32 (0.11–1.0) | 0.050* | 0.39 (0.11–1.41) | 0.149 |
| TST and/or IGRA Result, n = 253 | | | | 0.020* | | 0.069 |
| Positive | 98/114 (86%) | 16/114 (14%) | 2.39 (1.25–4.55) | 0.008* | 2.51 (1.15–5.49) | 0.021* |
| Negative | 100/139 (72%) | 39/139 (28%) | 1.00 (Ref) | | 1.00 (Ref) | |
| Not applicable | 44/53 (83%) | 9/53 (17%) | 1.91 (0.85–4.27) | 0.117 | 1.41 (0.53–3.80) | 0.494 |

Notes.
*Denotes statistical significance.

with LTBI offered treatment was clearly lower in the current study than reported in other studies, while the proportion of HCWs accepting treatment lay within the range of other study findings, suggests there is a service gap. Sixty-seven percent (198/297) of respondents

were in favour of being offered preventive TB treatment if they personally had evidence of LTBI.

It is of interest in this context that staff working in respiratory medicine showed significantly lower support for treatment of LTBI in HCWs in general (22%, 6/27) as well as for them personally (41%, 11/27). More scepticism towards treatment of LTBI among respiratory staff may be the result of increased awareness about potential side effects of preventive therapy with isoniazid, in particular the risk of drug-induced hepatitis. A US study found that the perception that LTBI treatment was harmful was an important barrier to HCWs adherence to work site TB screening and treatment policies (*Joseph et al., 2004*).

The reason why a large proportion (47%) of respondents with a positive test for LTBI felt that the question about the offer of preventive TB treatment did not apply to them was likely because the majority would not have had an encounter with a TB clinician to discuss preventive TB treatment, but would just have undergone the screening tests. It is possible that the question was not specific enough. We believe, however, that the high proportion of health care workers with evidence of LTBI who thought that the question about the offer of preventive TB treatment was not relevant to them, indicates lack of awareness about the option of preventive treatment among those who have likely been infected with TB.

Previous research in the current study setting indicated that physicians' decisions on treatment of LTBI were based on individualised appraisals of risks and benefits of LTBI treatment, and that there was a bias against offering treatment to people born overseas, men and HCWs (*Dobler, Luu & Marks, 2013*). We can only speculate about the reasons why HCWs with evidence of LTBI would not be targeted more frequently for preventive TB treatment. If HCWs are seen as part of routine HCW screening (rather than as part of a contact tracing investigation), the time of infection in HCWs with LTBI is usually unclear and the risk of TB reactivation is not as high as in recent TB infection. Further, HCWs may be perceived to have an ongoing risk of TB exposure at the workplace negating the benefits of preventive TB treatment. Importantly, there is evidence that physicians are reluctant to receive preventive TB treatment themselves, (*Bhanot et al., 2012*) possibly explaining why they would be less likely to offer it to other HCWs. A Canadian study suggested that physicians and patients may be more likely to initiate treatment if they trusted that a positive TST was not just simply the result of a previous BCG vaccination, but reflected true LTBI infection (*Xu & Schwartzman, 2010*).

The low rate of LTBI treatment among health care workers with evidence of LTBI in our study raises the question of the purpose of LTBI screening. The American Thoracic Society (ATS) and the Centers for Disease Control and Prevention (CDC) released a statement on LTBI screening and treatment in the year 2000 that endorsed the principle of "intention to test is intention to treat." In this statement screening for LTBI was only supported for persons at high risk for developing TB (persons with presumed recent *M. tuberculosis* infection and persons with clinical conditions associated with progression to active TB) who would be treated for LTBI if they had a positive screening test (*American Thoracic Society, 2000*). In many countries, including Australia, however, screening for LTBI is routinely performed among all healthcare workers (or at least in those with a significant stay in a TB endemic setting), even if they are not considered to be at a high risk of developing TB.
Thus, the "intention to test is intention to treat" principle does not necessarily apply to health care workers. In Australia, the primary purpose of healthcare worker screening is to exclude active TB (on chest X-ray) in persons who have a positive test for LTBI and to get a "baseline" result for LTBI, which can be helpful for assessing the risk of infection in case of future work-related TB exposure. The finding that 73% of respondents in our survey supported that Australian born health care workers without significant overseas stay should be screened for LTBI can possibly be explained in this context.

More than one in five respondents indicated that they did not know the difference between active and latent TB. The lack of perceived knowledge was more prevalent among nurses than doctors and in those aged 30 years or less. Insufficient knowledge about TB in younger HCWs could possibly be a result of the relatively low incidence of TB in Australia since the 1970s or a consequence of changed education and information associated with TB screening in HCWs. Only 33% of staff (75/230) felt adequately informed about the meaning of their TST and/or IGRA result indicating a need for better explanation of screening test results.

A limitation of the study was that the survey was performed in a single hospital. However, a study conducted in contacts of patients with active TB in all of New South Wales showed that only 9.5% of TB contacts with evidence of LTBI received preventive TB treatment, indicating that low proportions of treatment for LTBI are likely a systemic problem in New South Wales and possibly beyond, rather than an issue confined to a single centre (*Dobler, 2013*; *Dobler & Marks, 2013*). As with any survey study, participant selection bias could not be excluded. As only 24% of all questionnaire recipients responded, it is possible that those with a positive test for LTBI were overrepresented, as they could have been expected to have had a stronger interest in the topic of TB and LTBI than those with negative test results. The respondents with evidence of LTBI were, however, were likely representative of health care workers at the hospital in general in terms of previous treatment experience for LTBI. A previous case-control study in the same setting indicated that only 17% (16/92) of health care workers with evidence of LTBI received treatment (*Dobler, Luu & Marks, 2013*), which was not significantly different from the 13% (14/106) found in the current study ($p = 0.48$).

A further limitation was the small number of HCWs who had been offered preventive TB treatment ($n = 14$), of whom only 9 had accepted treatment. These numbers were too small to show significant associations between these outcomes and HCWs characteristics.

## CONCLUSIONS

The majority of hospital staff had positive attitudes towards preventive TB treatment, but the proportion of staff with evidence of LTBI who had been offered preventive TB treatment was low. Perceived knowledge about LTBI was insufficient, especially among HCWs aged 30 years or less and nurses. Only one third of HCWs who had undergone testing for LTBI felt adequately informed about the meaning of their test results, indicating a need for better information.

This study has identified a gap between the demand for preventive TB treatment and the current proportion of hospital staff with LTBI offered preventive treatment. This

indicates the need for a more pro-active and consumer-orientated approach regarding LTBI in HCWs. We suggest that HCWs with evidence of LTBI should routinely be offered preventive TB treatment unless there is clear concern that the risks may outweigh the benefits.

The study also identified room for improvement in TB health literacy, particularly, for improved education about key TB concepts, such as the difference between LTBI and active TB and for education about the meaning of screening test results. There is evidence that the use of information leaflets can improve knowledge about TB (*Roy et al., 2011*). By improving the care provided to HCWs with LTBI it may be possible not only to reduce active TB infections in this group, but also for the vulnerable people they care for.

### Funding
The authors received no funding for this work.

### Competing Interests
The authors declare there are no competing interests.

### Author Contributions
- Vidya Pathak performed the experiments, analyzed the data, wrote the paper, prepared figures and/or tables, reviewed drafts of the paper.
- Zinta Harrington conceived and designed the experiments, reviewed drafts of the paper.
- Claudia C. Dobler conceived and designed the experiments, analyzed the data, contributed reagents/materials/analysis tools, wrote the paper, prepared figures and/or tables, reviewed drafts of the paper.

### Human Ethics
The following information was supplied relating to ethical approvals (i.e., approving body and any reference numbers):

South Western Sydney Local Health District Ethics Committee (project number 12/265).

### Data Availability
Raw data has been uploaded as Supplemental Information.

### Supplemental Information
Supplemental information for this article can be found online at http://dx.doi.org/10.7717/peerj.1738#supplemental-information.

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
