# Peer review of "Attitudes towards preventive tuberculosis treatment among hospital staff"

_PeerJ, doi:10.7717/peerj.1738_

## Round 0.1 · original submission · Minor Revisions

Please provide a point by point response stating how and where in the revised manuscript each issue raised by reviewers was addressed.

·

Basic reporting

The reporting of this data meet the standards outlined by the journal.
To summarize this study reports the results of a survey of attitudes towards latent tuberculosis screening amongst healthcare workers at a single centre. The results are analysed by comparing the answers according to subject demographics and previous experience. This is an important question as attitudes and personal experience of staff managing LTBI will have direct impact on screening processes.
The most significant finding is the difference in attitude between respiratory staff and other specialties, this is striking as this group has the most experience of treating latent TB. Another interesting finding is that most staff who tested positive for latent TB did not believe they were offered preventative TB therapy.

Experimental design

The survey had a low but not expected response rate, the response rate seemed similar between doctors and nurses. It would be interesting to know if other data was collected about all of the subjects that were offered a survey. Additionally it would be useful to have some statistical analysis as to whether the final sample was representative of the group offered the survey as a whole, or how it reflects the general composition of staff at this hospital.
Some of the questions in the survey may have been too ambiguous such that a large proportion of the subjects who tested positive for LTBI (47%) were able to answer NA to the question of whether they were offered TB preventative treatment - this is disappointing as it was of great interest to know whether these people were even offered treatment or not, however this cannot be corrected at this stage.
I could not find the results from survey question 28 in either the tables or text.
The data from the survey were statistically analysed appropriately.

Validity of the findings

Most of the findings were reported clearly and appropriately. The determination of whether subjects knew the difference was based on asking subjects if they 'felt confident' that they knew the difference rather than an objective test and I think the discussion should reflect this, as this complex concept is often misunderstood.

Additional comments

Some minor typographic or poor grammar errors include line 87 “chest xray is done in case of a positive TST”, and line 184 “only specialty had” and should be rephrased. A reference for the statement in line 258 would be useful.
Table 2 would be clearer if the specialty was listed after the profession and the BCG and TST findings listed at the end.
The adage that ‘one should only test those intended for treatment’ does not appear applied by the respiratory specialists in this data set, this raises the question of whether respiratory specialist had been involved in developing the LTBI testing policy for this hospital, this may be beyond the scope of this study but the results of question 28 would be pertinent here with some comment.

Reviewer 2 ·

Basic reporting

Major comments:
- line 100: you say that doctor and nurses were randomly approached and asked to fill out questionnaires. Could you say more about this? You may know that people actually can’t randomly select things, in the true sense of randomness. There will always be a pattern, or a bias. That’s why we use tools such as a list of random numbers, a random number generator, or at least, picking a random starting point and then approaching every Xth person. Additionally, how did you know that one of these people hadn’t already completed a questionnaire? Or that they might later fill out the questionnaire that came with their pay slips (although it’s the rare person who willingly fills out a questionnaire twice, unless they’re getting a financial reward).
- lines 154 and 155: You say 295 of 311 respondents had undergone screening, and then report the percent who were screened at the hospital’s TB clinic. The denominator of the second percent is 284. Why isn’t it 295 – the number of people who had undergone screening?
- various lines in Results and in the Tables: as soon as I read “Attitudes towards preventive TB treatment” in Results and then looked at Table 2, I was confused. The N’s and percents did not match. I came to realize that you dropped people from Table 2 who had no opinion. And in fact, a couple sentences after you start giving these results, you provide a parenthetical comment that these people were dropped in the analysis presented in the Table. This is not acceptable – you cannot have a mismatch between text and table. You should choose one approach and be consistent.

Minor comments:
- line 60: you don’t provide a year for this WHO cite, either here in the text or in the References list
- line 102: you say “had” twice
- line 132: you make a common mistake in labeling your regression “multivariate.” That term actually refers to analyses with more than one outcome variable. “Multivariable” is the term you should use. See Bertha Hidalgo and Melody Goodman. Multivariate or Multivariable Regression? Am J Public Health. 2013 January; 103(1): 39–40.
- line 174: I think you meant to say “The majority of respondents supported the idea that hospital staff…”
- line 207: women were also less likely to know the difference, to a substantial degree. I assume you didn’t mention that only because it wasn’t borne out in the adjusted analysis.
- line 231: “A couple of…” is too colloquial
- line 254: interesting that your previous study found a bias against treating people born overseas. In the US, physicians are more likely to treat the foreign-born, viewing them as outsiders, and less likely to treat the US-born, unless they have other risk factors like homelessness or drug abuse.

Experimental design

"no comments"

Validity of the findings

I have done similar work around knowledge and attitudes toward Latent TB treatment so I was very interested to read your paper. Focusing on health care workers is of particular importance, both because my and other people’s studies have shown that they are more reluctant to take LTBI treatment, and because of the possible threat to their patients if they develop active TB. As shown in your Figure 1, it’s scary that 116 people tested positive and only 14 were offered treatment. And your overall finding of less interest in LTBI treatment among respiratory staff is also of great concern.

---

## Round 0.2 · accepted · Accept

The Reviewers issues have now been adequately addressed.

·

Basic reporting

All points have been clarified.

Experimental design

All queries have been responded to adequately.

Validity of the findings

Revised manuscript addresses all previous queries well.

Additional comments

This manuscript addresses an interesting question and makes a useful contribution to the field.